# Products of Bisphenol A Degradation Induce Cytotoxicity in Human Erythrocytes (In Vitro)

**DOI:** 10.3390/ijms24010492

**Published:** 2022-12-28

**Authors:** Katerina Makarova, Ewa Olchowik-Grabarek, Krzysztof Drabikowski, Justyna Kurkowiak, Katarzyna Zawada

**Affiliations:** 1Department of Organic and Physical Chemistry, Faculty of Pharmacy, The Medical University of Warsaw, Banacha 1, 02-097 Warsaw, Poland; 2Laboratory of Molecular Biophysics, Department of Microbiology and Biotechnology, Faculty of Biology, University of Bialystok, Konstanty Ciolkowskiego 1J, 15-245 Bialystok, Poland; 3Laboratory of Biological Chemistry of Metal Ions, Institute of Biochemistry and Biophysics Polish Academy of Sciences, Pawińskiego 5a, 02-106 Warsaw, Poland

**Keywords:** BPA, Fenton, erythrocytes

## Abstract

The aim of this work has been to study the possible degradation path of BPA under the Fenton reaction, namely to determine the energetically favorable intermediate products and to compare the cytotoxicity of BPA and its intermediate products of degradation. The DFT calculations of the Gibbs free energy at M06-2X/6-311G(d,p) level of theory showed that the formation of hydroquinone was the most energetically favorable path in a water environment. To explore the cytotoxicity the erythrocytes were incubated with BPA and three intermediate products of its degradation, i.e., phenol, hydroquinone and 4-isopropylphenol, in the concentrations 5–200 μg/mL, for 1, 4 and 24 h. BPA induced the strongest hemolytic changes in erythrocytes, followed by hydroquinone, phenol and 4-isopropylphenol. In the presence of hydroquinone, the highest level of RONS was observed, whereas BPA had the weakest effect on RONS generation. In addition, hydroquinone decreased the level of GSH the most. Generally, our results suggest that a preferable BPA degradation path under a Fenton reaction should be controlled in order to avoid the formation of hydroquinone. This is applicable to the degradation of BPA during waste water treatment and during chemical degradation in sea water.

## 1. Introduction

Bisphenol A (BPA) is widely used as a monomer or additive in polymer production. Such polymers are present in many products, e.g., food packaging, PVC, thermal receipts [1], dental sealants etc. [2]. The production of BPA was above 5 million tons in 2015 and it is estimated to rich 10 million tons in 2022 [3]. It is not surprising that BPA is detected in water effluents, soil leachates, drinking water and various home products [3,4]. The main routes of human exposure to BPA are through the consumption of the food packed in BPA containing cans and bottles, as well as dermal exposure and inhalation of household dust [3,5]. Thus, BPA’s presence is repeatedly found in the human breast milk (0.0004 to 21 µg/mL) [6,7,8], urine [7,9,10] and blood (0.40 ng/mL) [11].

Multiple studies report BPA’s adverse effects on human health, such as lower fertility, increased risk of cancer [12,13], development of metabolic [14,15] and cardiovascular diseases [15]. Among the suggested mechanisms of BPA action are binding to the estrogen-related receptor gamma and the oxidative stress [16,17].

That is why there is a constant search for the new methods of BPA degradation. The most efficient is degradation based on the reaction of BPA with the hydroxyl radical, i.e., the Fenton reaction. Generally, a Fenton reaction occurs when H_2_O_2_ reacts with Fe^2+^ and as a result the hydroxyl radicals are generated. It is suggested that the same reaction could be responsible for BPA degradation (chemical one) in seawater where both H_2_O_2_ and Fe^2+^ are present [18].

The BPA chemical degradation in seawater is reported to take up to 60 days [18]. On the other hand, the high efficiency of the BPA degradation (above 90%) during the short time (9 min) was reported for the photo-Fenton reaction and the ultrasound assisted Fenton reaction, etc. [19,20]. However, it was also shown that after the initial BPA degradation there is a number of intermediate degradation products present for up to 48 h. Katsumata at al. [21] suggested a scheme of BPA degradation in the presence of OH radical with 6 intermediate products of BPA degradation, including phenol, hydroquinone and 4-isopropylphenol. However, there is a lack of information for which the reaction path is energetically favorable.

Previously, we studied the xenoestrogenic properties of these intermediate products of the BPA degradation on the zebrafish embryo model [22]. We showed that the 4-isopropylphenol had the highest toxicity to the zebrafish embryos, followed by BPA, phenol and hydroquinone when compared at the same concentration. Otolithes abnormalities induced by 4-isopropylphenol and BPA were caused by the binding of these compounds to the estrogen-related receptor gamma. However, such morphological changes as cardiac edema, low heartbeat rate and delay in development in zebrafish embryos could not be explained by the xenoestrogenic activity of BPA and its products. Other studies suggest that the oxidative stress induced by BPA could be involved [23]. Indeed, BPA and phenol were shown to induce oxidative stress in human erythrocytes as well as lead to the hemolysis [24,25]. Additionally, it was reported that phenol and hydroquinone induced hemolysis in *Dicentrarchus labrax* erythrocytes [26], but still there is a lack of data for human erythrocytes and also there is a lack of studies for the 4-isopropylphenol.

In this work, we have determined the most favorable BPA degradation path under the Fenton reaction with the DFT calculation and ESR spin trapping. This path led to the formation of phenol, hydroquinone and 4-isopropylphenol. Then, we compared the cytotoxicity on human erythrocytes of BPA and three products of its degradation. The results of erythrocytes studies were confirmed by standard screening on nematode *C. elegans*. The energy description of the BPA degradation under the Fenton reaction could be used to control the degradation reaction and lead it to the formation of the least cytotoxic intermediate product. The same energy description could be used in further studies of BPA chemical degradation in seawater. The information about RONS generation induced by hydroquinone and 4-isopropylphenol could be used to explain the oxidative stress-induced morphological changes in zebrafish or other model organisms

## 2. Results

### 2.1. Spin Trapping ESR

The various BPA degradation methods are based on the effective production of hydroxyl radicals. In this work, we have used standard Fenton reaction to generate hydroxyl radicals and FDMPO spin trap to monitor the reaction. In the Fenton reaction, in the absence of BPA only FDMPO/OH radical adduct was observed with aN = 13.7 G and aF = 2.6 G, whereas in the presence of BPA the additional radical adduct with aN = 13.7 G and aF = 2.7 G appeared (Figure 1a). We could suggest that this additional radical adduct is a result of BPA degradation. Moreover, the integrated ESR signal intensity of FDMPO adduct in the Fenton reaction in the presence of BPA is higher than the one observed in the “pure” Fenton reaction. This is not only due to the generation of the additional radical adduct, FDMPO/X, but also due to the higher OH production (Figure 1b). This indicates that the degradation of BPA leads to the formation of a new radical adduct as well as increases the generation of OH radicals in comparison to the initial Fenton reaction.

### 2.2. DFT Calculations of Degradation Scheme

The degradation scheme of BPA under the Fenton reaction has been suggested by Katsumata et al. [21]. We adopted the degradation scheme for the DFT calculations i.e., we included additional step of extraction of a hydrogen atom from a water molecule by a hydroxyphenyl radical (Figure 2). Generally, the results of DFT calculations are strongly dependent on the level of theory and a solvent model (Table 1). However, there are similar trends in the values independent from the basis set except for the reaction 6, where at the MO62X level of theory the Gibbs free energy in water has a rather high positive value (50.16 kcal/mol) vs. negative values (−74.93, −84.7 and −64.12 kcal/mol) calculated for MO62X in gas and B3LYP in gas and in water. Based on the Gibbs free energy of the reaction the most energetically favorable reactions in water solvent are **1**, **2** and **4**.

### 2.3. Hemolysis

Bisphenol A as well as such products of its degradation as phenol, hydroquinone and 4-isopropylphenol caused significant hemolytic changes in erythrocytes after 1, 4 and 24 h of incubation (Figure 3). The hemolysis of human erythrocytes increased with the concentration of the studied compound and the time of incubation. After 1 h incubation, 4-isopropylphenol at 5 μg/mL and BPA and phenol at 10 μg/mL caused a statistically significant increase in the percentage of hemolyzed erythrocytes, whereas hydroquinone induced a statistically significant increase in the degree of hemolysis only at high concentration range (100–200 μg/mL) (Figure 3A). When compounds are compared at 200 μg/mL concentration, the highest increase in degree of hemolysis was induced by BPA (2.35 ± 0.20%) followed by phenol (1.68 ± 0.04%) and 4-isopropylphenol (1.58 ± 0.09%). The smallest effect was observed for hydroquinone (0.067 ± 0.3%).

We observed a statistically significant increase in the degree of hemolysis after 4 h of incubation of the erythrocytes with BPA, 4-isopropylphenol and phenol at 5 μg/mL. On the other hand, for hydroquinone the statistically significant increase in hemolysis of erythrocytes after 4 h incubation was observed in the range of concentration from 40 to 200 μg/mL (Figure 3B). Again, at 200 μg/mL concentration BPA led to the highest increase in the degree of hemolysis of the erythrocytes (5.64 ± 0.37%), whereas hydroquinone, 4-isopropylphenol and phenol induced similar changes (3.80 ± 0.23%, 3.76 ± 0.11% and 3.09 ± 0.11%, correspondently). After 24h of incubation, a statistically significant increase in the degree of hemolysis was noted for BPA and phenol at 10–200 μg/mL and for hydroquinone and 4-isopropylphenol at 20–300 μg/mL (Figure 3C). The highest increase of hemolysis was induced by BPA (36.26 ± 1.39%), followed by hydroquinone (26.15 ± 0.80%), phenol (21.06 ± 0.27%) and 4-isopropylphenol (16.58 ± 0.29%).

### 2.4. Content of the Reduced Glutathione (GSH)

We observed a statistically significant decrease in GSH percentage after 1 h of incubation with hydroquinone, phenol and 4-isopropylphenol at 5 μg/mL and with BPA at 10 μg/mL (Figure 4A).

Hydroquinone decreased the GSH percentage most strongly (to 16.19 ± 1.14% at concentration of hydroquinone 200 μg/mL), whereas BPA, phenol and hydroquinone at the same concentration reduced the GSH content to a similar level (to 75.48 ± 1.17%, 71.87 ± 1.00% and 66.51 ± 0.72%, correspondently). After 4 h of incubation all compounds induced a statistically significant decrease in GSH percentage at only 5 μg/mL (Figure 4B). Hydroquinone showed the strongest effect on GSH (the final percentage of 5.27 ± 0.23%), followed by 4-isopropylphenol (56.70 ± 1.75%), phenol (61.05 ± 1.38%) and BPA (65.40 ± 0.63%). The effect was dose dependent.

### 2.5. Reactive Oxygen and Nitrogen Species (RONS)

Hydroquinone and 4-isopropylphenol generated a statistically significant level of RONS at 10 μg/mL, whereas BPA and phenol induced a statistically significant level of RONS at 20 μg/mL (Figure 5). The highest level of RONS was observed for hydroquinone (970.00 ± 4.08%) at 200 μg/mL concentration, followed by 4-isopropylphenol (548.49 ± 3.44%) and phenol (290.50 ± 4.23%), and the effect was dose-dependent. The increase of RONS induced by BPA with increasing concentration of BPA was low (from 100.00 ± 1.67% to 118.31 ± 1.39%).

## 3. Discussion

The main source of BPA contamination in aquatic environment are BPA production factories and leakage from plastic debris [27,28]. It is suggested that BPA degrades in natural water, although different mechanisms are responsible for BPA degradation in river and sea waters [18]. In river waters BPA is usually degraded very fast (half-life 3–6 days at 25 °C) by bacteria [29,30], though it was reported that the degradation could take over 90 days [31]. Various bacterial strains exhibit the ability to metabolize BPA, e.g., those belonging to *Sphingomonas* sp., *Pseudomonas* sp. or *Novosphingobium* sp. [32]. It was shown that the dominating metabolites were hydroquinone, 4-hydroxybenzaldehyde, 4-hydroxybenzoic acid and 4-hydroxyacetophenone [33,34,35]. The mechanisms of BPA biodegradation have been discussed in detail in a review by Noszczyńska and Piotrowska-Seget [35]. BPA degradation in river waters in the presence of ROS is considered to be via formation of hydroquinone, i.e., similar to metabolism in vivo.

On the contrary, in sea water no correlation with the count of bacteria was observed and therefore chemical degradation mechanism of BPA is suggested [18]. This chemical mechanism is based on ROS, especially hydroxyl radicals formed in the presence of iron i.e., the same Fenton-like reaction used during waste water treatment. However, the chemical degradation of BPA in sea water is much slower (in the study of Kocaman et al. [31] the BPA concentration in natural seawater has not changed significantly throughout the 150 days of the experiment) than typical waste water treatment, where BPA degrades in 9 min [21]. Moreover, various factors could inhibit BPA degradation under Fenton-like reactions [36]. For this reason, basic description of BPA degradation under the Fenton-like reaction is needed.

The work has aimed to study the possible degradation paths of BPA under the Fenton reaction and to determine the energetically favorable intermediate products. In this work, we used the scheme of BPA degradation proposed by Katsumata et al. [21]. Since it has been shown previously that phenyl radicals could obtain H from H_2_O [37,38], we have modified these steps of reactions for DFT calculations (reactions 3 and 5) to be with H_2_O instead of H to make it energetically correct. There have been some previous attempts to monitor BPA degradation under Fenton reaction with spin trapping ESR [36]. Authors used DMPO spin trap; they observed the decrease of DMPO/OH signal intensity and suggested that OH radical was reacting with BPA. However, only a single spectrum at one moment of the reaction was shown without following the reaction for a longer time. In addition, DMPO/OH is known to a have short lifetime (several minutes) so it is hard to analyze the reaction path. Therefore, we used the FDMPO spin trap to monitor the BPA degradation under the Fenton reaction, since it forms very stable FDMPO/OH radicals adducts with a lifetime of up to several days. We observed the increase of the ESR signal intensity of FDMPO/OH adduct in the Fenton reaction in the presence of BPA. That could indicate that the possible route of BPA under the Fenton reaction is through reaction 3 or 5 (Figure 1), where phenol or 4-isopropylphenol intermediate products could be formed. As was shown previously the M06-2X/6-311G(d,p) level of theory is more accurate for this type of compounds, that is why we discuss only these results and present results of B3LYP/6-311G(d,p) solely as the reference to other studies [39]. Based on the DFT calculations at M06-2X/6-311G(d,p) level of theory using PCM model for water, we could suggest that reactions 4 and 2, leading to the formation of hydroquinone, are the most energetically favorable route for BPA degradation. Furthermore, reaction 6, leading to the formation of 4-(2-hydroxy)isopropylphenol could be possible as different values are obtained for MO62X in water and for other levels of the theory. The trapped radical (FDMPO/X) is rather an O-centered radical, as C-centered radical adducts would give higher nitrogen hyperfine splitting [40]. It is probably formed at the subsequent steps of the reaction as both radicals (Radical A and Radical B) are C-centered radicals. Of course, it could be that other radicals are also formed, however, they are not trapped by FDMPO. Thus, another more selective spin trap should be used if more detailed information about reaction path is required.

It is suggested that BPA might be highly soluble in body fluids and thus accumulates in saliva and blood [41]. Erythrocytes, which play an important role in various biochemical and physiological processes, are the most abundant blood cells. Thus, they are significantly more exposed to the action of BPA and products of its degradation [24]. Hence, as the next step, we compared the cytotoxicity of three intermediate products of BPA degradation, i.e., phenol, hydroquinone and 4-isopropylphenol, on human erythrocytes. The hemolysis in the presence of BPA and phenols has been studied previously [24,25], however, there is no complete data for hydroquinone and there is a lack of information on 4-isopropylphenol. It was shown that BPA induced hydroxyl radicals generation and led to lipid peroxidation [42]. The suggested mechanism is that BPA crosses the lipid membrane, then binds to hemoglobin iron and causes dissociation of the hemoglobin subunits. As a result, the released iron acts as a pro-oxidant and induces the generation of hydroxyl radicals with subsequent lipid peroxidation and hemolysis.

However, it was shown that lipid peroxidation is not very important for phenol induced hemolysis, which is rather based on the damage of heme proteins and other proteins [25]. In another study, it was shown that hydroquinone induced 77% hemolysis in *Dicentrarchus labrax* erythrocytes after 24 h of incubation at 20 °C, whereas in phenol treated cells there were no significant differences with control [26]. It is suggested that autooxidation of hydroquinone leads to the formation of free radicals, which leads to cell membrane damage and further hemolysis, similar to BPA [43]. No mechanism for 4-isopropylphenol-induced hemolysis was suggested. Our results indicate that the hemolysis degree in BPA-treated human erythrocytes was the largest one, followed by the one in the presence of hydroquinone, phenol and 4-isopropylphenol. The difference in phenol-induced hemolysis in our study and one presented by Boge et al. [26] could be attributed to the nature of *Dicentrarchus labrax* erythrocytes.

The suggested mechanism of free radical generation by BPA, phenol and hydroquinone, which leads to hemolysis could, also, be the cause of the decrease in the level of GSH. It was shown previously that several phenolic compounds may change the activities of enzymes participating in oxidative stress [25]. The living cell’s defensive mechanism against free radicals could be affected by the damage of such enzymes as CAT and SOD. The decrease in GSH level in the presence of phenol could be attributed to the glutathione interactions with this compound and the formation of glutathione conjugate [25]. Hydroquinone was also reported to deplete the level of GSH and to reduce the myeloperoxidase and the catalase activities, but no interactions with SOD were reported [44]. In our studies, phenol and 4-isopropylphenol similarly decreased the GSH level, whereas the strongest effect was observed for hydroquinone. That could be due to the highest level of RONS generated by hydroquinone. In the previous studies [44] hydroquinone was also shown to generate high a level of free radicals. Based on a high level of RONS generation by 4-isopropylphenol as well as a similar level of GSH depletion we could suggest that it induces hemolysis in a similar way to hydroquinone or phenol.

To verify the higher toxicity of hydroquinone and 4-isopropylphenol than this of BPA, we used *C. elegans* as in vivo model (Appendix A). In the range of studied concentrations, none of the compounds caused oxidative stress (Appendix A), however 4-isopropylphenol caused some neurotoxicity (Appendix A). Previously, we showed that 4-isopropylphenol caused the highest toxicity to zebrafish embryos [22]. Such difference in results on erythrocytes, *C. elegans* and *D. rerio* suggest that mechanism of toxicity/oxidative stress of BPA and products of its degradation depends on the complexity of the organism. 

## 4. Materials and Methods

### 4.1. Chemicals

Bisphenol A (BPA), phenol and 4-isopropylphenol were purchased from Sigma-Aldrich (Poznan, Poland). Hydroquinone, hydrogen peroxide (30%), ferrous sulfate (FeSO_4_·7H_2_O) and all other chemicals were supplied by Avantor Performance Materials Poland S.A. (Gliwice, Poland). All reagents were of analytical grade and were used as received without any further purification. Test solutions were freshly prepared for each independent experiment by dissolving BPA, phenol, 4-isopropylphenol and hydroquinone in a buffer (at pH 7.0, in ultrapure water).

### 4.2. DFT Calculations

All DFT calculations were performed in Gaussian 09 program. The structures (Figure 1) were optimized at B3LYP/6-31G(d,p) level of theory separately for gas and water. The water was accounted for using the PCM model [45]. The enthalpy and the Gibbs free energy of the reaction were calculated according to “Thermochemistry in Gaussian” at B3LYP/6-311G(d,p) and MO6X/6-311G(d,p) levels of theory in gas and water [46].

### 4.3. Electron Spin Resonance (ESR)

The ESR measurements were performed on a MiniScope MS 200 spectrometer from Magnettech at room temperature (24 ± 1 °C) using 50 µL capillary tubes. Typical instrument settings were, sweep width 69 G, microwave power 10 mW, modulation amplitude 0.5 G, and sweep time 20 s.

### 4.4. Spin Trapping ESR

Hydroxyl radicals were generated using the Fenton reaction. FDMPO solution was prepared using PBS buffer (pH 7.5). 20 µL of 10 mM FDMPO solution was mixed with 20 µL of 20 mg/L BPA or water (for reference measurements). Then, 20 µL of 0.5 mM FeSO_4_ solution and 20 µL of 0.5 mM H_2_O_2_ solution were added. The sample was vortexed for 30 s. The first spectrum was recorded after 2 min from the addition of the H_2_O_2_ solution. The subsequent spectra were recorded automatically every 60 s.

### 4.5. ESR Spectra Simulation

All spectra simulations were performed in MATLAB using the Easyspin toolbox [47]. The “Garlic” function was used to simulate fast isotropic spectrum of FDMPO spin adducts. Optimization of the simulated spectra was performed using the Simplex method.

### 4.6. Erythrocyte Isolation

The blood of 12 healthy volunteers (aged 20–50) was obtained from Regional Blood Donation and Blood Treatment Center in Bialystok, Poland. Human blood was collected in tubes containing 18 mg EDTA/10 mL as an anticoagulant. Blood was centrifuged (600× *g*, 15 min, 4 °C). Then, plasma layer and leukocytes were removed. Erythrocytes were washed three times with 0.9% NaCl. Then hematocrit was determined in the resulting erythrocytes mass.

### 4.7. Reactive Oxygen and Nitrogen Species Determination

2′,7′-Dichlorofluorescin diacetate (DCFH-DA) was incubated in 3 mM NaOH for 15 min prior to its addition to erythrocytes. Erythrocytes (10% suspension in PBS, pH = 7.4) with 40 µM DCFH-DA were incubated for 30 min at 37 °C. The cells were washed twice with PBS (1500× *g*, 5 min) and 2 mL of BPS buffer was added. Then, 0.5% of erythrocytes with DCFH-DA suspension was incubated for 1 h with all the studied compounds (10–200 µg/mL) and 4 h with BPA (10–200 µg/mL). The changes in DCF fluorescence intensity were registered at λ = 480 nm excitation wavelength and λ = 530 nm emission wavelength [48].

### 4.8. Level of the Reduced Glutathione (GSH)

A total of 1 mL of 1% suspension of erythrocytes in PBS (pH 7.4) was incubated with the studied compounds (10–200 µg/mL) for 1 and 4 h at 37 °C. Then, 200 µL of 25% trichloroacetic acid was added and the samples were centrifuged (400× *g*, 10 min). In total, 0.5 mL of 0.5 M phosphate buffer (pH 7.8) and 0.05 mL of Ellman’s reagent (5 mM) were added to 0.5 mL of the supernatant. Then, samples were incubated for 30 min at room temperature in total darkness. The absorption was measured spectrophotometrically at 414 nm [25].

### 4.9. Hemolysis

A total of 1 mL of 1% erythrocytes suspension in buffer (150 mM NaCl, 10 mM Tris-HCl, pH = 7.4) was incubated with bisphenol A, hydroquinone, phenol and 4-isopropylphenol (10–200 µg/mL) for 1, 4 and 24 h at 37 °C. In total, 4 mL of buffer (150 mM NaCl, 10 mM Tris-HCl, pH = 7.4) was added after incubation to samples with studied compounds. 4 mL of water was added to the control sample. The erythrocytes were centrifuged (400× *g*, 15 min). The absorbance in supernatant was measured at λ = 540 nm.

The hemoglobin absorbance in the control sample after complete hemolysis of the erythrocytes with water was taken as 100%. Percent of hemolysis of the erythrocytes in all samples was calculated relatively to the control sample with water. (1)$$H[\%]=\frac{A}{A_{100\%}}\cdot 100\%$$
where *H* [%]—percent of hemolysis; *A*—hemoglobin absorbance in the samples with the erythrocytes incubated with tested compounds; *A*_100%_—haemoglobin absorbance in the sample after complete hemolysis of the erythrocytes with water (100%) [24].

### 4.10. Statistical Analysis

Statistical analyses were performed using StatSoft, Inc. (2014). STATISTICA (data analysis software system), version 12. The statistically significant difference between groups was checked using one-way ANOVA or the non-parametric equivalent—Kruskal–Wallis test. The assumption of normal distribution was analyzed using the Shapiro–Wilk test and the assumption of homogeneity of variance by the Brown–Forsythe test. To compare groups to the control Dunnett’s post-hoc test was applied. All computations were applied at significance level of 0.05.

## 5. Conclusions

Based on DFT calculations the most energetically favorable degradation path of BPA is via the formation of a hydroquinone intermediate product. On the other hand, in the spin trapping studies, i.e., during the BPA degradation under the Fenton reaction, we observed the increase of the FDMPO/OH radical adducts production and the formation of the additional FDMPO/X radical adduct.

All studied compounds induced hemolysis in erythrocytes, however, the mechanism of BPA induced hemolysis is rather different from the one shown by phenol, hydroquinone and 4-isopropylphenol. BPA generated the lowest level of RONS, and the effect of concentration was very small, whereas other compounds generated higher levels of RONS, and the effects were dose dependent. In addition, the largest decrease in GSH level was induced by hydroquinone, the strongest pro-oxidant in the set of studied compounds.

To conclude, the waste water treatment process or the chemical degradation in sea waters could lead to the formation of phenol or 4-isopropylphenol intermediate products which are less cytotoxic than BPA and hydroquinone. However, this process should be controlled to lead to phenol only as our and other studies reported toxicity and high xenoestrogenic properties of 4-isopropylphenol.

## Figures and Tables

**Figure 1 ijms-24-00492-f001:**
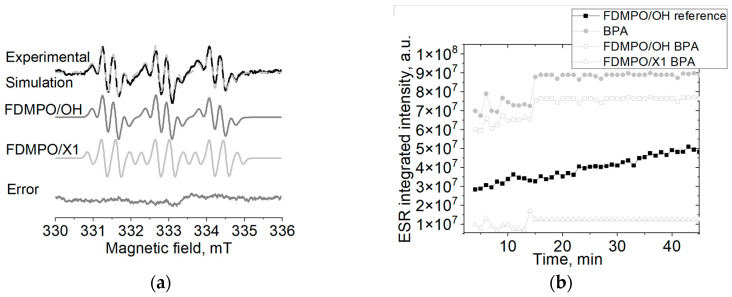
(**a**) ESR spectra of FDMPO radicals adducts formed in the presence of BPA under Fenton reaction; (**b**) Kinetics of the integrated intensity of ESR spectra from FDMPO radical adducts formed in the presence of BPA under Fenton reaction.

**Figure 2 ijms-24-00492-f002:**
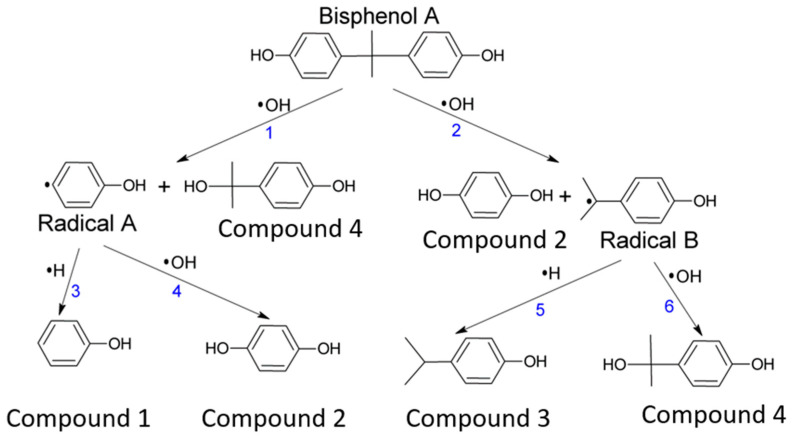
BPA degradation scheme in the presence of OH radicals adopted from Katsumata et al. [21]. Phenol (compound **1**), hydroquinone (compound **2**), 4-isopropylphenol (compound **3**), 4-(2-hydroxy)isopropylphenol (compound **4**).

**Figure 3 ijms-24-00492-f003:**
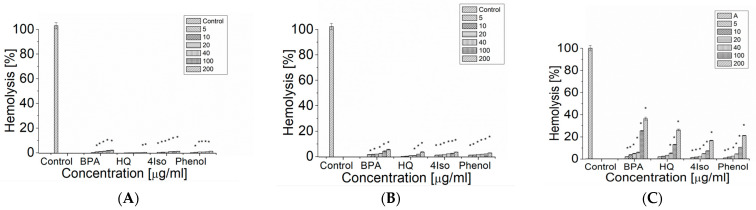
The hemolytic changes in the control erythrocytes and in those incubated with BPA, hydroquinone, phenol and 4-isopropylphenol in the concentration range 5–200 µg/mL after 1 h (**A**), 4 h (**B**) and 24 h (**C**) of incubation. The data are presented as the means ± SE. * the effects of compounds were statistically significant.

**Figure 4 ijms-24-00492-f004:**
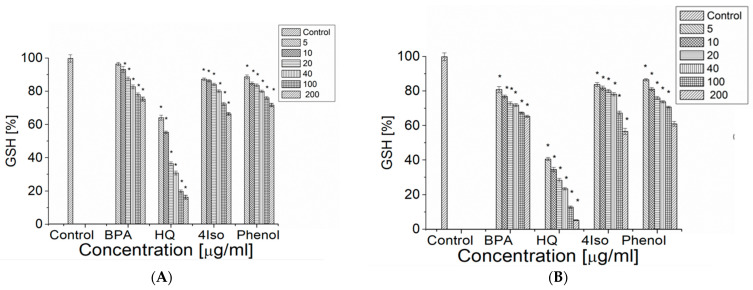
The level of reduced glutathione (GSH) in the control erythrocytes and the ones incubated with BPA, hydroquinone, phenol and 4-isopropylphenol in the concentration range 5–200 µg/mL after 4 h (**A**) and 24 h (**B**) of incubation. The data are presented as the means ± SE. * the effects of compounds were statistically significant.

**Figure 5 ijms-24-00492-f005:**
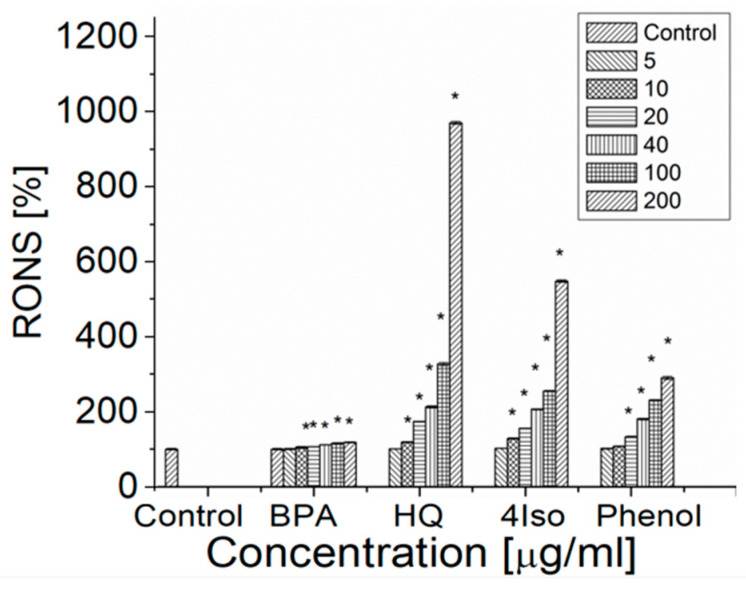
The level of RONS generated in the control erythrocytes and the erythrocytes incubated with BPA, hydroquinone, phenol and 4-isopropylphenol in the concentration range 5–200 µg/mL after 24 h of incubation. The data are presented as the means ± SE. * the effects of compounds were statistically significant.

**Table 1 ijms-24-00492-t001:** Gibbs free energy and enthalpy of the BPA degradation under the Fenton reaction according to Figure 1 calculated at M06-2X/B3LYP/6-311G(d,p) levels of energy in gas and in water solvent (PCM model).

	Enthalpy (ΔH) [kcal/mol]				Gibbs Free Energy (ΔG) [kcal/mol]			
	B3LYP/6-311G(d,p)		M06-2X/6-311G(d,p)		B3LYP/6-311G(d,p)		M06-2X/6-311G(d,p)	
	Gas	Water	Gas	Water	Gas	Water	Gas	Water
1	−23.87	−4.09	1.10	1.17	−27.23	−7.34	−2.29	−2.14
2	−53.67	−34.06	−23.75	−147.99	−57.88	−38.19	−28.08	−152.49
3	27.82	10.10	2.24	3.70	28.95	11.23	3.35	4.81
4	−126.84	−106.42	−112.32	−111.74	−115.36	−94.98	−100.73	−100.19
5	57.03	39.38	28.10	153.78	58.49	40.84	29.63	155.52
6	−97.04	−76.45	87.47	37.42	−84.70	−64.13	−74.93	50.16

## Data Availability

All data presented in this study are available in the main body of the manuscript.

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
