# Peer review of "Products of Bisphenol A Degradation Induce Cytotoxicity in Human Erythrocytes (In Vitro)"

_ijms, 2022, doi:10.3390/ijms24010492_

Round 1

Reviewer 1 Report

The authors have presented data on degradation products of BPA and their effects on erythrocytes including the effect of BPA itself.

This is a critical problem given the massive sources of BPA that leads to contamination of sea water and river water. 

The knowledge that BPA degradation can be controlled in a way that less toxic intermediates are formed, is interesting and useful. However, in my opinion it may be good if the authors discuss the biological degradation pathways in detail and rationalize why they picked the three metabolites for their study. 

Further, just RBC lysis does not provide a concrete basis for cytotoxic effects. Using cell culture like gastrointestinal, muscle, lung or liver cell line may add robustness to the claim of cytotoxicity.

I would be happy to see at least two human cell line data.

Author Response

Dear Reviewer,

thank you for your comments.  We believe that we addressed all of them and corrected the article properly

"The knowledge that BPA degradation can be controlled in a way that less toxic intermediates are formed, is interesting and useful. However, in my opinion, it may be good if the authors discuss the biological degradation pathways in detail and rationalize why they picked the three metabolites for their study"

We have added details about biological degradation pathways to the discussion: " In river waters BPA is usually degraded very fast (half-life 3-6 days at 25°C) by bacteria [29,30], though it was reported that the degradation could take over 90 days [31]. Various bacterial strains exhibit the ability to metabolize BPA, e.g. these belonging to Sphingomonas sp., Pseudomonas sp. or Novosphingobium sp. [32]. It was shown that the dominating metabolites were hydroquinone, 4-hydroxybenzaldehyde, 4-hydroxybenzoic acid and 4-hydroxyacetophenone [33-35]. The mechanisms of BPA biodegradation have been discussed in detail in a review by NoszczyÅ„ska and Piotrowska-Seget [35]."

Hydroquinone is a by-product in both wastewater treatment (Fenton) and biological degradation of BPA in rivers. Phenol and 4-isoprophylpenol is only reported to appear in Fenton degradation, which could also occur during the degradation in sea waters. We concentrated on Fenton reaction products, as this reaction could be controlled during the wastewater treatment, whereas we do not have any control over the biological degradation process in rivers. Still, our results could be used to describe the degradation of BPA in sea waters. (This is discussed in the article).

2. "Further, just RBC lysis does not provide a concrete basis for cytotoxic effects. Using cell culture like gastrointestinal, muscle, lung or liver cell line may add robustness to the claim of cytotoxicity.

I would be happy to see at least two human cell line data."

Due to the short revision time (only 10 days) and the fact, that we do not have access to the cell cultures, we did additional experiments on N2 worms (C. elegans) as this is a well-established toxicity model for in vivo studies. The results on N2 worms we added to the supplementary materials.  They confirm the toxicity of the 4-isoprophyphenol, similarly to our previous studies with zebrafish emrbyos.

Reviewer 2 Report

Please see the attached word file.

Author Response

Dear Reviewer,

thank you for your comments. All suggested corrections were taken into account.

Reviewer 3 Report

It is an interesting study concerning the possible degradation path of BPA under Fenton and a preliminary screening of the induced cytotoxicity of its intermediate products. The results clearly presented and are fully supported by the results. The only drawback is the quality of the presentation of the figures. The resolution of the graphs must be improved, the font size must be increased and all the figures must have similar font, font size, etc.

Author Response

Dear Reviewer,

thank you for your comments.  Inndeed, the font size and font style we not appropriate. We corrected all figures to have the same font ( Arial) and font size ( 48 for axis titles and numbers). 

Round 2

Reviewer 1 Report

The authors have addressed the concern adequately. 

Author Response

Dear reviewer,

we have added new results on C.elegans model i.e. screening oxidative stress induces in C.elegans by bisphenol, phenol, 4-isoprophylphenol and hydroquinone treatment. Indeed , the studied compounds act differently in different systems. 

The following comments was added to the main article "

To verify the higher toxicity of hydroquinone and 4-isopropylphenol than this of BPA, we used C.elegans as in vivo model (Supplementary materials, Fig. S1,S2). In the range of studied concentrations, none of the compounds caused oxidative stress (Fig.S1), however 4-isopropylphenol caused some neurotoxicity. (Fig. S2, C). Previously, we showed that 4-isopropylphenol caused the highest toxicity to zebrafish embryos. [22] Such differences in results on erythrocytes, C.elegans, and D.rerio suggest that the mechanism of toxicity/oxidative stress of BPA and products of its degradation depends on the organism's complexity."